

# Hadron structure from basis light-front quantization

**Chandan Mondal**[1,2,*], **Jiangshan Lan**[1,2,3], **Kaiyu Fu**[1,2], **Siqi Xu**[1,2],
**Zhi Hu**[1,2], **Xingbo Zhao**[1,2], **James P. Vary**[4]
**(BLFQ Collaboration)**

**1** Institute of Modern Physics, Chinese Academy of Sciences, Lanzhou 730000, China
**2** School of Nuclear Science and Technology, University of Chinese Academy of Sciences,
Beijing 100049, China
**3** Lanzhou University, Lanzhou 730000, China
**4** Department of Physics and Astronomy, Iowa State University, Ames, IA 50011, U.S.A.

* mondal@impcas.ac.cn

## Abstract

**We present our recent progress in applying basis light-front quantization approach to investigate the structure of the light mesons and the nucleon.**

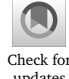
## 1 Introduction

The structures of hadronic bound states are embedded in the light-front wavefunctions (LFWFs) obtainable as the solutions of the eigenvalue equation of the Hamiltonian: $P^- P^+ |\Psi\rangle = M_h^2 |\Psi\rangle$, where $P^{\pm} = P^0 \pm P^3$ represents the light-front (LF) Hamitonian ($P^-$) and the longitudinal momentum ($P^+$) of the system, respectively, with the mass squared eigenvalue $M_h^2$. Basis light-front quantization (BLFQ) [1] provides a computational framework for solving relativistic many-body bound state structure in quantum field theories. In this paper, we report our study on the structures of the pion and the nucleon from the BLFQ approach.

## 2 Light mesons

With quarks and gluons being the explicit degrees of freedom for the strong interaction, we consider an effective LF Hamiltonian which incorporates LF QCD interactions [2] relevant to constituent quark-antiquark and quark-antiquark-gluon Fock components of the mesons with a complementary three-dimensional (3D) confinement [3]. We adopt the LF Hamiltonian

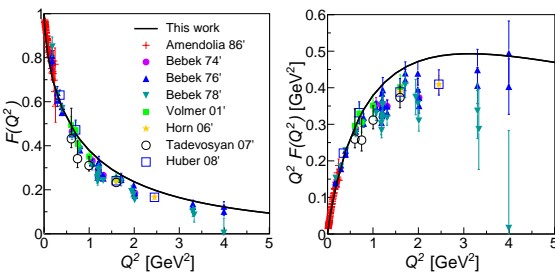

Figure 1: The EMFF of the pion. The experimental data can be found in Ref. [4].

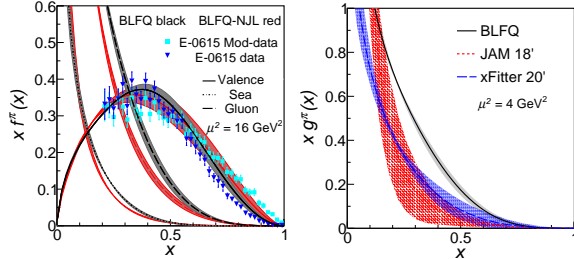

Figure 2: Left: the PDFs of the pion at $\mu^2 = 16$ GeV$^2$. Right: the pion's gluon PDF at $\mu^2 = 4$ GeV$^2$ is compared with the global fits, JAM [9] and xFitter [10].

$P^- = P^-_{\text{QCD}} + P^-_C$, where $P^-_{\text{QCD}}$ and $P^-_C$ are the LF QCD Hamiltonian and a model for the confining interaction. With one dynamical gluon in the LF gauge

$$P^-_{\text{QCD}} = \int d^2 x^\perp dx^- \left\{ \frac{1}{2} \bar{\psi} \gamma^+ \frac{m_0^2 + (i\partial^\perp)^2}{i\partial^+} \psi - \frac{1}{2} A_a^i \left[ m_g^2 + (i\partial^\perp)^2 \right] A_a^i \right.$$
$$\left. + g_s \bar{\psi} \gamma_\mu T^a A_a^\mu \psi + \frac{1}{2} g_s^2 \bar{\psi} \gamma^+ T^a \psi \frac{1}{(i\partial^+)^2} \bar{\psi} \gamma^+ T^a \psi \right\}, \tag{1}$$

where $\psi$ and $A^\mu$ are the quark and gluon fields, respectively. Meanwhile, the confinement in the leading Fock sector is given by [3]

$$P^-_C P^+ = \kappa^4 \left\{ x(1-x) \vec{r}_\perp^2 - \frac{\partial_x[x(1-x)\partial_x]}{(m_q + m_{\bar{q}})^2} \right\}, \tag{2}$$

with $\kappa$ being the strength of the confinement.

With the framework of BLFQ [1], we solve the Hamiltonian in the leading two Fock sectors. We truncate the infinite basis space by introducing limits $K$ and $N_{\text{max}}$ in longitudinal and transverse directions, respectively. By fitting the constituent parton masses and coupling constants as the model parameters, we obtain a good quality description of light meson mass spectroscopy [4]. We evaluate the pion electromagnetic form factor (EMFF) and the parton distribution functions (PDFs) from our resulting LFWFs obtained as eigenfunctions of the Hamiltonian. The LFWFs are boost invariant in the longitudinal and the transverse directions. The BLFQ approach employs a suite of analytical and numerical techniques for setting up and solving the eigenvalue problem in a convenient basis space [3–5]. Complementary insights into nonperturbative QCD can be achieved from the discretized space-time euclidean lattice [6] and the Dyson-Schwinger equations of QCD [7].

The EMFF of the charged pion is compared to the experimental data in Fig. 1. We find consistency between our results and the precise low $Q^2$ EMFF data. Meanwhile, notable devi-

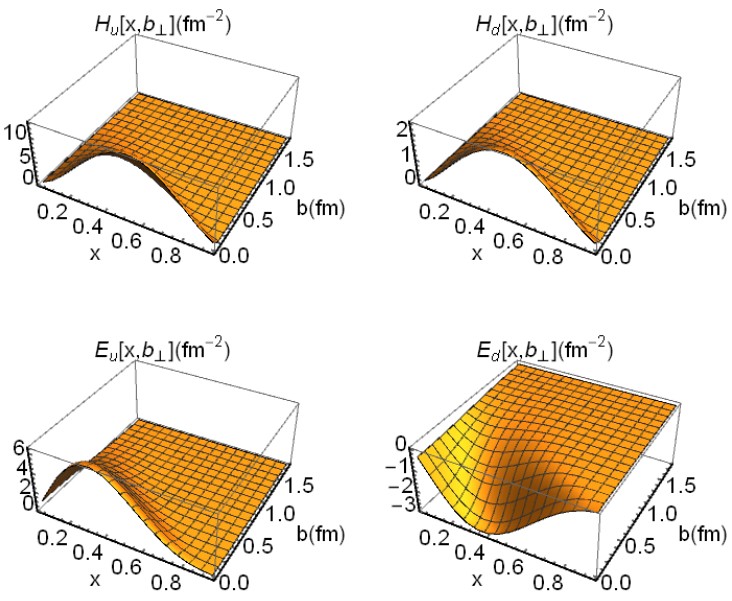

Figure 3: The quark unpolarized GPDs as functions of $x$ and $b_\perp(\equiv b)$ for the up (left panel) and down quark (right panel) in the proton.

ations have been observed at large $Q^2$. Note that our choice of $N_{\max}$, implies the UV regulator $\Lambda_{\mathrm{UV}} \sim b\sqrt{N_{\max}} \approx 1$ GeV [4]. Thus, our predictions are most reliable in the low $Q^2$ region.

We show the pion PDFs after QCD evolution in Fig. 2. Our prediction for the pion valence quark distribution is found to be consistent with the reanalyzed FNAL-E-615 data. The gluon distribution significantly increases in our approach compared to that in the BLFQ-NJL model [5,8] as well as to the global fits [9,10] as can be seen from Fig. 2. We notice that gluons carry $\{39.5, 42.1, 43.9, 44.6, 45.1\}\%$ of pion momentum at the scale $\mu^2 = \{1.69, 4, 10, 16, 27\}$ GeV$^2$, respectively.

## 3 Nucleon

We solve for the mass eigenstates for the nucleon from a light-front effective Hamiltonian in the leading Fock sector representation, consisting of a 3D confinement (pairwise), Eq. (2), and a one-gluon exchange interaction with fixed coupling. [11, 12]. The parameters in the effective interaction are fixed by fitting the nucleon mass and the flavor Dirac form factors. We then employ the resulting LFWFs to calculate the EMFFs, PDFs, GPDs, TMDs and many other observables of the nucleon [11–13].

The unpolarized GPDs for zero skewness, $H(x, b_\perp)$ and $E(x, b_\perp)$, as functions of $x$ and the transverse impact parameter $b_\perp$ for up and down quarks are shown in Fig. 3. The GPDs evaluated in our BLFQ approach are qualitatively similar to the GPDs calculated in phenomenological models [14, 15].

The quark TMDs: $f_1(x, p_\perp^2)$, $g_{1L}(x, p_\perp^2)$, and $h_1(x, p_\perp^2)$ in the proton are presented in Fig. 4. The qualitative behavior of our results is also consistent with the TMDs calculated in phenomenological models [16, 17].

Fig. 5 shows our results for the valence quark PDFs of the proton, where we compare the valence quark distributions after QCD evolution with the global fits by different Collaborations

and the measured data from COMPASS Collaboration. Our unpolarized valence PDFs for both up and down quarks are found to be in good agreement with the global fits. Meanwhile, we find that the spin dependent PDFs ($g_1$ and $h_1$) for the down quark agree well with measured data (or the global fits). However, for the up quark, our spin dependent distributions tend to overestimate the data below $x \approx 0.3$.

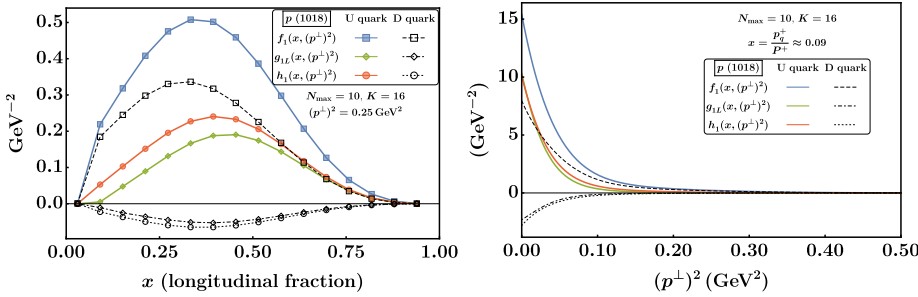

Figure 4: The quark TMDs: $f_1(x, p_\perp^2)$, $g_{1L}(x, p_\perp^2)$, and $h_1(x, p_\perp^2)$ (related to leading twist PDFs) in the proton. Left: vs $x$ for fixed $p_\perp^2$; right: vs $p_\perp^2$ for fixed $x$ [13].

# 4 Conclusion and outlook

BLFQ has been proposed as a nonperturbative framework for solving quantum field theory. We have reported the structure of the pion and the nucleon from the BLFQ appraoch. The LFWFs obtained as the eigenvectors of the light-front QCD Hamiltonian for the light mesons by considering them within the constituent quark-antiquark and the quark-antiquark-gluon Fock spaces employed to produce the pion EMFF and the PDFs. For the nucleon, we begin with an effective light-front Hamiltonian incorporating confinement and one gluon exchange interaction for the valence quarks suitable for low-resolution properties. The LFWFs obtained as the eigenvectors of this Hamiltonian were then used to generate the proton PDFs, GPDs, and TMDs.

The resulting LFWFs can be employed to study other parton distributions, such as the generalized TMDs, the Wigner distributions, the double parton correlations as well as the mechanical properties etc. On the other hand, this work for the mesons can be extended to higher Fock sectors to incorporate, for example, sea degrees of freedom as well, while for the nucleon this can be extended to investigate gluon and sea quarks distributions.

# Acknowledgements

C. M. thanks the Chinese Academy of Sciences Presidents International Fellowship Initiative for the support via Grants No. 2021PM0023. X. Z. is supported by new faculty startup funding by the Institute of Modern Physics, Chinese Academy of Sciences, by Key Research Program of Frontier Sciences, Chinese Academy of Sciences, Grant No. ZDB-SLY-7020, by the Natural Science Foundation of Gansu Province, China, Grant No. 20JR10RA067 and by the Strategic Priority Research Program of the Chinese Academy of Sciences, Grant No. XDB34000000. J. P. V. is supported in part by the Department of Energy under Grants No. DE-FG02-87ER40371, and No. DE-SC0018223 (SciDAC4/NUCLEI).

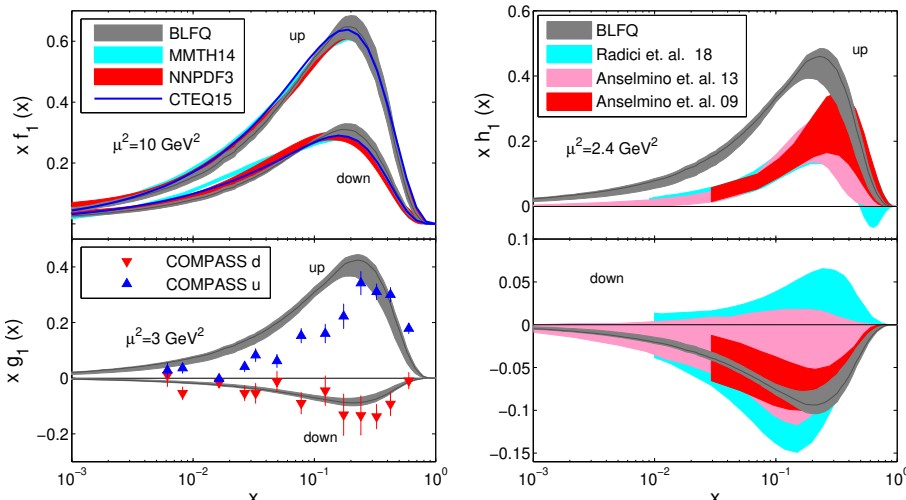

Figure 5: The leading twist quark PDFs: $f_1(x)$, $g_1(x)$ (left panel) and $h_1(x)$ (right panel) in the proton [11, 12].

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
