# Peer review of "Hadron structure from basis light-front quantization"

_SciPost Physics Proceedings, doi:SciPost Phys. Proc. 10, 036 (2022)_

## Round 1 · Referee Report · Anonymous (Referee 1) · 2022-3-29

Strengths

1. A coherent and compact summary of some interesting work, with many applications in hadron structure.

2. Mostly comprehensible, even to a non-expert.

Weaknesses

1. A little context of why LFWF are a good/interesting strategy, and what alternative approaches exist would be useful for general readers.

2. The EMFF description is derscribed as "impressive"; there are, however, notable deviations from the data, with the LH plot systematically overshooting higher-Q2 points. Quantitatively, how good is the fit; is the high-Q2 mismatch understood; and how does this description look from other approaches?

3. The nucleon description shows plots of GPDs and TMDs, but not the collinear PDFs, where comparison to global-fit models would be interesting. Could such a plot be included?

4. A short conclusion mentioning the next intended steps from here would be nice to have, and conclude the contribution well.

Report

The criteria are generally met for publication, but a few small improvements (as mentioned in the weaknesses section) would make it much better.

Requested changes

See weaknesses list

  • validity: high
  • significance: high
  • originality: high
  • clarity: high
  • formatting: excellent
  • grammar: good

Author:  Chandan Mondal  on 2022-04-11  [id 2371]

(in reply to Report 1 on 2022-03-29)
Category:
answer to question

I. A little context of why LFWF are a good/interesting strategy, and what alternative approaches exist would be useful for general readers.

Our response: We thank the referee for the suggestions. To implement the referee’s suggestion, we have added the following sentences at the end of the paragraph just after Eq. (2): →

“The LFWFs are boost invariant in the longitudinal and the transverse directions. The BLFQ approach employs a suite of analytical and numerical techniques for setting up and solving the eigenvalue problem in a convenient basis space [3-5]. Complementary insights into nonperturbative QCD can be achieved from the discretized space-time euclidean lattice [6] and the Dyson-Schwinger equations of QCD [7].”

II. Page 4: The EMFF description is derscribed as ”impressive”; there are, however, notable deviations from the data, with the LH plot systematically overshooting higher-Q^2 points. Quantitatively, how good is the fit; is the high-Q^2 mismatch understood; and how does this description look from other approaches ?

Our response: We adjusted the model parameters by fitting the light mesons mass-spectroscopy. Note that we did not fit the EMFF of the pion. This is our prediction. The notable deviation of our EMFF from the data at high-Q 2 can be understood from the basis truncation in the transverse direction (Nmax ) in our BLFQ approach. Our current truncation (Nmax = 14) implies the UV regulator ≈ 1 GeV, where b is the harmonic oscillator scale parameter. Thus, our predictions are most reliable in the low Q^2 region, where our result is also consistent with other theoretical approaches and phenomenological models (lattice QCD, Dyson-Schwinger equations, light-front holography, constituent quark model etc.).

Modification: We have modified the following sentence “We find an impressive agreement between our results and the precise low Q^2 EMFF data.” to

“We find consistency between our results and the precise low Q^2 EMFF data. Meanwhile, notable deviations have been observed at large Q 2 . Note that our choice of N max , implies the UV regulator ≈ 1 GeV [4]. Thus, our predictions are most reliable in the low Q ^2 region.”

III. The nucleon description shows plots of GPDs and TMDs, but not the collinear PDFs, where comparison to global-fit models would be interesting. Could such a plot be included?

Our response: As suggested by the referee, we have now included the plot for the nucleon PDFs in Fig. 5. The corresponding discussions have been added before conclusion.

IV. A short conclusion mentioning the next intended steps from here would be nice to have, and conclude the contribution well.

Our response: As suggested by the referee, a short conclusion and outlook have been included.

Attachment:

response_ISMD_de41iyQ.pdf

---

## Round 2 · Author Response

Dear Editor,
We thank the referee for their comments and suggestions. We address the referee’s points one-by-one and summarize the resulting changes we made to the manuscript.
Yours sincerely,
BLFQ Collaboration
We thank the referee for their comments and suggestions. We address the referee’s points one-by-one and summarize the resulting changes we made to the manuscript.
Yours sincerely,
BLFQ Collaboration

---

## Round 2 · List of Changes

I. Page 2, after Eq. (2), end of the paragraph, we have included the following text:
The LFWFs are boost invariant in the longitudinal and the transverse directions. The BLFQ approach employs a suite of analytical and numerical techniques for setting up and solving the eigenvalue problem in a convenient basis space [3-5]. Complementary insights into nonperturbative QCD can be achieved from the discretized space-time euclidean lattice [6] and the Dyson-Schwinger equations of QCD [7].
II. Page 2, second last paragraph: We have modified text in the following form:
We find consistency between our results and the precise low Q^2 EMFF data. Meanwhile, notable deviations have been observed at large Q^2. Note that our choice of Nmax, implies the UV regulator ΛUV ∼ b\sqrt{Nmax} ≈ 1 GeV [4]. Thus, our predictions are most reliable in the low Q^2 region.
III. Figure 5 has been include, the corresponding discussion has been added last paragraph before conclusion.
IV. Conclusion and outlook section has been included.
The LFWFs are boost invariant in the longitudinal and the transverse directions. The BLFQ approach employs a suite of analytical and numerical techniques for setting up and solving the eigenvalue problem in a convenient basis space [3-5]. Complementary insights into nonperturbative QCD can be achieved from the discretized space-time euclidean lattice [6] and the Dyson-Schwinger equations of QCD [7].
II. Page 2, second last paragraph: We have modified text in the following form:
We find consistency between our results and the precise low Q^2 EMFF data. Meanwhile, notable deviations have been observed at large Q^2. Note that our choice of Nmax, implies the UV regulator ΛUV ∼ b\sqrt{Nmax} ≈ 1 GeV [4]. Thus, our predictions are most reliable in the low Q^2 region.
III. Figure 5 has been include, the corresponding discussion has been added last paragraph before conclusion.
IV. Conclusion and outlook section has been included.

---

## Editorial Decision

published